# Brain Functional Correlates of Episodic Memory Using an Ecological Free Recall Task

**DOI:** 10.3390/brainsci11070911

**Published:** 2021-07-09

**Authors:** Francesco Neri, Stefano F. Cappa, Lucia Mencarelli, Davide Momi, Emiliano Santarnecchi, Simone Rossi

**Affiliations:** 1Siena Brain Investigation and Neuromodulation Lab (Si-BIN Lab), Unit of Neurology and Clinical Neurophysiology, Department of Medicine, Surgery and Neuroscience, University of Siena, 53100 Siena, SI, Italy; lucia.mencarelli92@gmail.com (L.M.); momi.davide89@gmail.com (D.M.); simone.rossi@unisi.it (S.R.); 2Institute for Advanced Study, IUSS, 27100 Pavia, PV, Italy; stefano.cappa@iusspavia.it; 3IRCCS Mondino Foundation, 27100 Pavia, PV, Italy; 4Berenson-Allen Center for Noninvasive Brain Stimulation, Beth Israel Deaconess Medical Center, Harvard Medical School, Boston, MA 02115, USA; Esantarn@bidmc.harvard.edu; 5Department of Cognitive Neurology, Harvard Medical School, Boston, MA 02115, USA

**Keywords:** fMRI, memory task, angular gyrus, brain networks, IPL, episodic memory

## Abstract

Episodic Memory (EM) allows us to revive a past event through mental time-travel. The neural correlates of memories recollection have been identified in hippocampal regions and multiple neocortical areas, but few neuroimaging studies have used an ecological task such as a free recall of a structured story. Using an ecological fMRI-free recall (FR) task, we aimed to investigate the relevant recruitment of the brain networks associated with the story recollection process and its performance. Fourteen healthy participants listened to a brief story and were tested for Immediate-Recall (IR), a task that is widely used in a neuropsychological evaluation. Then, the subjects underwent an fMRI session, where they had to perform a free recall (FR) of the story subvocally. Finally, the participants were tested for Delayed-Recall (DR). IR and DR scores were significantly (*r* = 0.942; *p* < 0.001) correlated. FR enhanced the activity of the Language, the Left Executive Control, the Default Mode and the Precuneus brain networks, with the strongest BOLD signal localized in the left Angular Gyrus (AG) (*p* < 0.05; FWE-corrected). Furthermore, the story recall performance covaried with specific network activation patterns and the recruitment of the left anterior/posterior AG correlated, respectively, with higher/lower performance scores (*p* > 0.05). FR seems to be a promising task to investigate ecologically the neural correlates of EM. Moreover, the recruitment of the anterior AG might be a marker for an optimal functioning of the recall process. Preliminary outcomes lay the foundation for the investigation of the brain networks in the healthy and pathological elderly population during FR.

## 1. Introduction

Episodic Memory (EM) is a Long-Term Memory (LTM) subsystem allowing storage and recollection of personally experienced past events with their contextual connotations (What Where and When or WWW) [1]. The multiple episodic traces are merged through a binding process dependent upon the medial temporal lobes [2]. EM is characterized by autonoetic consciousness, i.e., the sense of self-recollection associated with remembering personally lived events, at which one was present resulting in a “mental time travel,” where past experiences are relived in the present [1]. The integrity of EM is crucial for social and communicative behavior [3] and its deterioration, especially in free recall tasks, is a potential marker of Alzheimer Disease (AD) onset [4,5].

At the clinical and the experimental level, the assessment of EM is usually conducted by asking subjects to recall lists of words [6], pictures [7] or faces [8], i.e., single items in isolated contexts. The aforementioned paradigms have the advantage to be highly controllable in an experimental environment, but they have received some criticism [9], as increasing the control on a specific phenomenon may compromise transfer of the results to real life conditions [10,11].

On the other hand, the recall of a short story has a long tradition in clinical neuropsychology [12]. It originates from the “Babcock story” [13], a verbal memory measure in which participants read a brief story and then are asked to recall it immediately. The story is then read a second time, and delayed recall is collected after an interval of 20 min. Performance on prose rather than isolated words recall benefits from the semantic organization of the information, reducing the load on the self-generation of recall strategies requiring executive resources [14]. This is supported by the worse performance of patients with frontal lobe lesions [15] and executive dysfunction [16] for unrelated words lists than for semantically related information.

In neuroimaging studies, a free recall of a structured event has been rarely adopted as a protocol to investigate the neural basis of EM retrieval process. fMRI activity has been measured while participants encode or retrieve a TV show or a movie [17,18]. Other authors have preferred to use an autobiographical memory task that consists of retrieving specific events situated in time and space, [10,19] but it requires access to a subject’s personal general knowledge, such as their name, addresses and generic events [9].

The medial temporal lobe is the neural core of the EM, as indicated by the investigation of human amnesia [20,21], but neuroimaging studies have provided important evidences about the role of multiple cortical areas in EM processes, including the dorsolateral prefrontal cortex [22,23,24] in encoding information and the posterior parietal cortex (PPC) subareas during memory retrieval: the Angular Gyrus (AG), the Precuneus, the Retrosplenial Cortex (RC), the Inferior Parietal Lobule (IPL) and the Inferior Parietal Sulcus (IPS) are all recruited during information recall [25,26].

Traditionally, the PPC has been linked to spatial function [27,28]. Substantial evidence, coming from event-related potential (ERP) and neuroimaging studies, demonstrates that the PPC is recruited during a memory task [26,29,30,31,32,33]. Functional connections between bilateral PPC and hippocampal formation have been identified as part of a large memory network [34]. Computational modeling suggests that PPC is further characterized by the graded variation of long-range connectivity to temporal and inferior frontal areas [35].

Specifically, AG seems to be strongly linked to brain areas governing memory functions and might support the retrieval of details during an event recall [36]. Bilateral AG is recruited during a free and cued recall of autobiographical memories [10,19] and a recent quantitative Activation Likelihood Estimate analysis has revealed that these activations are similar to those found during a recognition task [37], thus confirming a pivotal role of the AG during the recall of long-term mnestic information, in particular in context integration while processing narratives. This arrangement is compatible with an identical buffering mechanism, resulting in different emerging functions [38]. Recently, a free recall of movie scenes has been used as an ecological task during an fMRI acquisition, to study neural correlates of the EM and revealing that high level cortical PPC areas are recruited during a real-life event recall [18].

Lesion evidence also supports the role of PPC in EM. A prominent metabolic dysfunction in the lateral/medial parts of the parietal cortex has been observed in the early stage of Alzheimer’s disease [39]. Greater damage to the right AG leads to a poorer memory performance on a complex associative retrieval [40] and patients with AG lesion are less able than healthy participants to recall the multifeatured sources of an encoded stimulus [41].

Recently, it has been shown that specific subareas of the AG contribute differently to episodic and semantic retrieval [42] and continuous Theta Burst Stimulation (TBS) delivered on the AG interferes with free recall in autobiographical memory task but not in a cued autobiographical memory task, and neither in a free or cued paired words task. These results led to the hypothesis that the ventral PPC is involved in the subjective experience of memory [43].

Using an ecological task, our aims were to detect brain activation during verbal recollection following an auditory encoding, identify which subregions of the ventral PPC or IPL are functionally recruited and verify if specific IPL areas or brain networks might particularly contribute to a better EM performance.

We investigated a group of young healthy participants recalling a short prose passage during an fMRI scanning session. The prose recall task may be considered to be more “ecological” than list learning, as it better reproduces a situation of the real life, such as reading a newspaper or listening to news and recalling the content after a delay to transfer the information.

## 2. Materials and Methods

The data that support the findings of this study are available from the corresponding author, F.N., upon reasonable request.

### 2.1. Participants

Fourteen healthy, fully right-handed participants (nine females and five males; mean age = 29 years; SD = 3.6) took part in the experiment. They reported no history of language disorders, perception deficits or neurological or psychiatric disorders. The investigation was carried out in accordance with the latest version of the Declaration of Helsinki, and the protocol was approved by the Local Ethical Committee. All participants signed an informed consent. 

### 2.2. Paradigm

Preliminarily, eight brief Italian prose passages were selected from a total of 24 used in a previous study [44] and edited to obtain standardized passages (30 concepts; words mean: 158.6 ± 5.8) Then, the prose passages were vocally recorded using the software Waves MaxxAudio Pro (Waves Audio Ltd.; Tel Aviv, Israel) (average duration of the recordings: 73.5 s ± 3.9).

The participants were seated in an armchair inside a soundproof room and listened to a randomly selected passage through headphones. After listening, the subjects retrieved the story with a classic immediate recall (IR) task, trying to retrieve as many details as possible.

In the following 20 min, the subjects underwent a fMRI block design session. The participants wore earplugs and were asked to lay still on the scanner-bed with their eyes open and to alternate covertly between two tasks every ~50 s (20 fMRI volumes): a free recall (FR) of the passage and a backward counting (BC) starting from 100 and subtracting 7 until the next task-shift. The shifts between the tasks were determined by an experimenter’s signal. Mind wandering was not allowed and participants had to perform the same task for the entire block. If they finished repeating the passage during FR or they reached 0 during the BC, they had to start the task again until the next experimenter’s signal. To avoid a schematization of the information due to multiple repetitions, we asked the participants to repeat the story as similar as possible as the moment of the IR. Afterward, the participants were taken back to the soundproof room and asked to recall the passage in a delayed recall (DR) task.

Lastly, subjects filled out an ad-hoc Likert-structured questionnaire (points from 1: very low; to 7: very high), to collect some indices that might have conditioned the fMRI FR performance. Participants were asked to rate: (i) performance similarity between FR and IR, (ii) performance similarity between FR and DR, (iii) FR ability, (iv) thinking interference, (v) fMRI noise interference, (vi) backward counting ability, (vii) tasks interference, (viii) shifting ability and (ix) visual imagination.

### 2.3. Behavioral Data Analysis

For each participant, we calculated the total number of recalled concepts of the IR and DR. The behavioral data were analyzed using the Statistical Package for Social Science (SPSS 16.0; IBM Corp., Chicago, IL, SUA, 2017), investigating the relationship between IR and DR with a correlation and a Linear Regression Analysis. We calculated the mean, standard deviation and frequency distribution of the questionnaire results.

### 2.4. MRI Data Acquisition and Preprocessing

MRI data were acquired by means of a Philips Intera MRI 1.5 T scanner at “Policlinico Le Scotte” in Siena (Siena, Italy). Structural images were obtained with a whole brain T1-weighted Fast Field Echo 1 mm^3^ sequence (TR/TE = 30/4.6 ms, field of view = 250 mm, matrix 256 × 256, flip angle = 30°, slice number = 150). Whole brain fMRI data (178 volumes; 33 axial slices) were registered via a T2 BOLD-sensitive multi-slice echo planar imaging (EPI) sequence (TR/TE = 2.5 s/32 ms; field of view = 22 cm; image matrix = 64 × 64; voxel size = 3.44 × 3.44 × 3.8 mm 3; flip angle = 75°).

fMRI data were preprocessed with a custom-made pipeline implemented in MATLAB (MathWorks Inc., Sherborn, MA, USA) with freely available scripts and analyzed using Statistical Parametric Mapping (SPM12; Wellcome Department of Imaging Neuroscience, London, UK; 2012). The first three volumes of every fMRI sequence were discarded for the steady-state magnetization period and equilibration of scanner signal [45]. A total of 175 volumes were extracted from the normalized functional image of each subject.

For each subject, EPI time series were corrected using fieldmaps regression [46], which were then removed from non-cerebral tissues and slice timed (interleaved ascending acquisition). Finally, fMRI images were realigned and resliced to the mean volume for head motion correction. Structural images were co-registered to the mean volume of the processed functional images, obtaining grey matter, white matter and CSF images. The resulting group-specific brain template was normalized and resampled in a 3 × 3 × 3 mm voxel size to functional images. In order to compare the BOLD activation of FR and BC conditions, the 175 fMRI volumes were separated clustered in two time series consisting of 95 volumes for FR and 80 volumes for BC.

### 2.5. fMRI Analysis

#### 2.5.1. Group Level Analysis

Data were analyzed with the general linear model (GLM): microtime resolution: 23; microtime onset: 11 and two regressors of interest: FR and BC. The model was estimated using standardized residuals from the GLM. To test BOLD activation differences, FR and BC images were compared with a *t*-test (FR > BC and BC < FR contrasts). The significance level was set to *p* < 0.05 familywise error-corrected, (FWE-corrected), considering a minimal cluster size of 50 adjacent voxels, to guarantee adequate control for false positives 

#### 2.5.2. Weighted Dice Coefficient (wDC) Analysis

To characterize the BOLD signal of the FR condition in a brain functional network profile, a qualitative overlap analysis and a quantitative wDC analysis [47] were conducted. Thus, the spatial similarity index between functional brain networks and the significant BOLD activation of the FR > BC contrast was explored. Story recall task execution required an auditory-verbal processing and we selected Shirer’s Resting-State Networks (RSNs) atlas [48] to conduct the overlap analysis, because it includes an optimal subdivision of the language network.

#### 2.5.3. IPL wDC Analysis

As we were interested to define which parts of the PPC were predominantly recruited in EM, we explored the role of ventral part of PPC (or IPL) during the FR task, together with the correlation between activations of IPL subregions with retrieval performance.

Previous cytoarchitectonic parcellation studies [49,50,51,52] divided the IPL in “PF” and “PG” areas [52], where “P” is the parietal cortex, “F” corresponds to the supramarginal gyrus (SMG) and “G” is the AG. Recently, five parts of the SMG/F (PF, PFcm, PFop, PFt and PFm, where “op”, “t”, “cm” and “m” are, respectively, the opercular part, the thin cortical ribbon, the columnar/magnocellular part and the magnocellular part) and two areas of the AG/G (PGa and PGp, where “a” is the anterior part and “p” is the posterior part of the AG) were identified [49,50,51].

We conducted a quantitative wDC analysis, looking to the spatial similarity index between IPL atlas parcellation by Caspers and colleagues [49] and the significant BOLD activation of the FR > BC contrast.

#### 2.5.4. Performance-Related Analysis

In order to explore the relationship between behavioral and fMRI data, the IR score was included as a covariate of interest in the FR > BC contrast, masking the contrast with the significant activation map derived from the group level analysis. To characterize spatial similarity patterns, we conducted a quantitative wDC analysis between the output map with: (i) the whole brain using Shirer’s RSNs and (ii) the ventral PPC using IPL parcellation [49].

#### 2.5.5. MVPA Analysis

To unveil subtle patterns of activation, we carried out a multivariate pattern analysis (MVPA) using CONN toolbox. We compared FR and BC tasks and performed the analysis by retaining two components; this choice was motivated by the assumption that the number of components should be equal to 10–20% of the sample size (here between 1.4 and 2.80). We show the MVPA results in the specific section Appendix A (see Appendix A).

## 3. Results

### 3.1. Behavioral Performance

Participants retrieved an average of 16.9 (SD = 5.3) and 16.8 (SD = 5.5) concepts in IR and DR, respectively (~56% of the possible elements that can be recalled). The correlation between the IR and DR performance was highly significant (*r* = 0.942; n = 14; *p* < 0.001). Moreover, a significant regression equation was found (F_(1,12)_ = 94.47, *p* <.001) with an R^2^ = 0.887 and IR was identified as significant predictor of DR performance (β = 0.942, *p* < 0.001).

### 3.2. Questionnaire Ratings

The score of performance similarity judgment between FR with IR and FR with DR was globally high across subjects (means = 5.43 and 5.79 respectively); hence, we can assume that the participants’ FR and IR/DR performances were nearly equivalent. Participants rated as low the interference between task and of intrusive external thoughts (means = 1.79 and 1, respectively). Furthermore, subjects rated highly their FR and shifting abilities (means = 5.79 and 5.43, respectively) and tended to mentally visualize the passage during FR (mean = 5.64). The variable detected as the most disturbing during fMRI tasks execution was the magnetic resonance noise (mean = 4); however, this noise was always present. In Appendix A, the mean, standard deviation, median and frequency distribution values of the participants’ questionnaire response are reported.

### 3.3. Neuroimaging Results

#### 3.3.1. Group Level Analysis

The FR > BC contrast showed a significant activation (*p* < 0.05; FWE-corrected) of the bilateral PPC (AG in particular) with left sided prevalence, left precuneus, left frontal inferior cortex (pars opercularis and pars triangularis), left fronto-orbital cortex, middle temporal gyrus, medial frontal cortex, superior frontal gyrus, frontal pole and cerebellum bilaterally (see Figure 1A and Table 1 for details). The qualitative pattern of the activation trended towards a left lateralization. For BC > FR activation, see Figure 1B and Appendix A for details.

#### 3.3.2. RSNs wDC Analysis

We explored the spatial similarity between the results of FR > BC contrast and the functional RSNs. The qualitative analysis revealed that FR task pattern mostly resembled the topographical distribution of the dorsal part of default mode network (dDMN) and of the language network (LANG) (Figure 2A). In addition, networks of the left executive control (LECN) and of the precuneus (PREC) showed a pattern of a high spatial similarity with the FR > BC activation map. Quantitative wDC confirmed the pattern, with the highest spatial similarity index for LANG, dDMN, LECN and PREC (Figure 2B).

#### 3.3.3. IPL wDC Analysis

The wDC analysis using the IPL atlas [49] demonstrated a higher spatial similarity index for the AG of both hemispheres and the left SMG (Figure 3).

#### 3.3.4. Performance-Related Analysis

A wDC analysis was conducted using the suprathreshold correlational maps (FR > BC; IR performance as covariate; *p* > 0.05). A better IR performance led to the recruitment of the LECN, LANG, DMN, PREC and AS RSNs. In contrast, a worse recall performance predominantly engaged the LANG network, excluding the PREC network and lowering the LECN recruitment (Figure 4A,B). Moreover, wDC analysis results, considering IPL parcellation, showed that higher and lower IR performances recruited, respectively, the left anterior and left posterior AG (Figure 4B,C).

## 4. Discussion

The main aims of the current study concerned: (i) using an ecological task during an fMRI acquisition, thus combining an old tool that is widely used in neuropsychological practice with a relatively new neuroimaging technique to investigate the memory recollection process; (ii) investigating IPL and RSNs activation patterns involved during a clinically used story recall episodic memory task; (iii) verifying the relationship between brain activation patterns and the recall performance within the RSNs and the IPL. We discuss the results point-by-point, with a particular focus on the innovation of the paradigm and the neuroimaging findings.

### 4.1. Free Recall of a Story during fMRI and Behavioral Results

Free recall paradigms during fMRI acquisition have been very seldom used, with the exception of a cued free recall for autobiographical information [10,19] and a free recall task using movie scenes [18]. As far as we know, the current paradigm with auditory/verbal encoding/recollection has never been used as of yet. Our aim was to continuously load the “recall cognitive function” during FR fMRI blocks acquisition using non-autobiographical memories and simulating an ecological situation, where a person has to recall a story in order to remember it as well as possible. Participants reported a high level of concentration during the fMRI recall task and were perfectly able to switch from BC to FR tasks. They also referred not to be distracted by any interference due to internal thoughts or from the loud fMRI noise. Lastly, participants highly rated the recall similarity between repetitions inside and outside the scanner.

The IR/DR performance (~56% of correct concepts recalled) is comparable to previous findings: in 1987, Barigazzi and colleagues collected normative data using a prose passage (174 words/51 concepts). In this case, a subsample of 10 healthy young subjects recalled on average the 51% of total concepts [53]. 

We found a strong, positive correlation between IR and DR, verging to perfect linearity, thus demonstrating that the information remained unaltered and stable in the verbal long-term memory storage (i.e., absence of oblivion). The finding of a perfect correlation is an uncommon outcome in the field of behavioral experiments, but it is not totally new as far as prose memory studies are concerned: an excellent positive correlation has already been found between immediate and delayed recall of prose passages [53,54], though our correlation was even greater, probably because subjects were able to review the information several times during the fMRI session, hence consolidating it. This is in agreement with the concept that rehearsal stabilizes the quantity and quality of the information [55]. However, it must be emphasized that subjects did not know that they would be retested after the end of the scanning session, so the high correlation between the two recalls remained relatively unexpected.

### 4.2. Brain Activation during Memory Recall

The BC > FR contrast activation was mostly localized in brain areas related to the dorsal attention network (DAN) (see Figure 1B, Appendix A and Appendix A), supporting previous fMRI results and contrasting BC task with a resting state condition [56]. Thus, it seems that BC activation is independent from the contrast, supporting the assumption that the significant BOLD signal differences of the FR > BC contrast reflected the pure execution of the EM task.

In the FR > BC contrast, the BOLD signal mainly overlapped with the LANG, the dDMN the LECN and the PREC networks [48]. As shown in Figure 1, the BOLD signal was mostly enhanced within the IPL area during the story recall. The outcome is in line with other EM findings, where a cued recall paradigm of autobiographical verbal-auditory information was used [10,42]. This brain hub is linked to language processing and, together with the left IFG, constitutes the largest part of the LANG network, which was recruited during the FR task. As highlighted from the wDC analysis, the increased IPL BOLD signal was strongly left lateralized in our right-handed subjects during the task. In particular, the BOLD activity in the anterior and posterior parts of the AG (PGa, PGp respectively) and in the posterior part of the SMG (PFm) were enhanced in the left hemisphere, whereas only the anterior and posterior parts of the AG (PGa, PGp) were recruited in the right hemisphere. 

Generally, the ventral IPL (comprising AG and SMG) is engaged for mnestic, attentional and linguistic tasks, both in the right and left hemisphere [28,57,58], characterizing the ventral IPL as a multimodal, non-specific, convergence node for multiple high cognitive components. In the Parietal Unified Connectivity-biased Computation (PUCC) model, the ventral IPL is conceptualized to be implicated in different cognitive tasks and the magnitude/localization of the activation in its subregions is dependent on the task performed by the participants [59,60]. A larger left lateralization enhancement is observed for linguistic recall tasks [61], possibly reflecting the attentional effort of the ventral IPL in retrieving verbal information, thereby integrating the attentive, linguistic and mnestic abilities required to perform the auditory task (auditory encoding + verbal recall).

Moreover, it has been hypothesized that the activation of the ventral IPL does not reflect the memory process itself, but more parsimoniously the bottom-up attentional function directing and monitoring the complex operations of information recall, conceptualized in the Attention-to-Memory (AtoM) model [57,58]. The phenomenon occurs whenever an information of interest consciously calls attentional resources [30]. In our paradigm, the task switching signal might direct the subject’s attentional resources towards the recall process, acting as a cue and activating the brain networks necessary and sufficient to perform the task. 

The present results indicate that the bottom-up attentional and memory–linguistic networks are not mutually exclusive during the recall of an information, but they can cooperate with each other to perform the task. 

### 4.3. Correlation of RSNs and IPL Patterns of Activation with Memory Performance

Exploiting the excellent correlation between IR and DR measures and the high ratings of similarity between FR and IR/DR reported in the questionnaire, we assumed the FR/IR/DR performance likeness and we used IR information to assess its correlation with the fMRI data. We found that the level of recall performance was associated to different patterns of RSNs and IPL recruitment (Figure 4). 

Subjects with a good IR performance tended to have a balanced activation pattern, mostly localized within LANG, dDMN and LECN RSNs and, although to a lesser extent, within PREC and AS functional networks. Instead, a worse IR performance led to a null activation of the PREC and a strong decrease of the LECN network with a contemporary enhancement of the LANG network activation, while no changes for the dDMN recruitment were observed. 

Usually, LECN is engaged during working memory tasks with a high performance outcome [62], although recent studies demonstrated a conjoint activation of the dDMN and LECN during the execution of an internal focus task [63]. During FR, dDMN was activated regardless of the details recalled, thus reflecting the network activation in response to the general mnestic demand, requiring the direction of the attentional resources towards the internal information representation, through a bottom-up mechanism triggered by the experimenter’s shift-task signal. This outcome is in line with a recent finding, connecting the recruitment of the dDMN to the free recall process of movie scenes and highlighting how this brain network activation is similar across all subjects during the retrieval process [18]. Instead, the IR/FR performance was linked to different balance patterns of multiple recruitments of the LECN, dDMN and LANG and is dependent on the way how these brain networks simultaneously activated. 

Indeed, a better recalling performance was connected to a consistent and equable activation of the LECN dDMN LANG and, to a lesser extent, an activation of the PREC and AS RSNs. The role of the PREC during a recall seems pivotal and its activation is positively correlated with a better performance [64]. In the case of a poor recall performance, the networks balancing between LECN, dDMN and LANG decayed, causing an increase of the LANG and a decrease of the LECN recruitment; moreover, the activation of the PREC and the AS disappeared. In this case, we hypothesized that the subjects were trying to recall the words and concepts of the story, being unable to direct the necessary attentional resources inward to properly recall the stored information. These results agree with the type of task used: various RSNs must be activated simultaneously in order to have a good performance, requiring a contemporary involvement of linguistic (LANG), mnestic (PREC) attentional (dDMN, AS) and executive control (LECN) abilities.

Regarding the IPL activation, we observed two patterns within the AG. The right anterior AG was recruited more than its posterior part, independently of the memory performance. In the case of a better recall, the left anterior AG was more activated than its ipsilateral posterior part; in contrast, with a poor recall performance, we observed an opposite pattern: the left posterior was more activated than its anterior part. The increased activation of the left IPL has already been reported [65,66], but our results originally suggest a microscale segregation within the IPL based on performance in a free recall task. This evidence may open new investigations: non-invasive brain stimulation techniques might be used to perturb different subregions of the AG, in order to assess specific performance changes in EM tasks.

It is noteworthy that the posterior part of the angular gyrus (PGp) is functionally and structurally connected with the hippocampal structures [67] and part of a large scale memory network [34]. We hypothesize that the greater recruitment of the PGp, concomitant with a worse performance, may reflect a greater cognitive effort to recall information not completely stable in the long-term memory storage. 

### 4.4. Limitations and Future Directions

Although this study provided some interesting results, it is worth mentioning some limitations that could serve as useful tips for future investigations. The main limitation is given by the sample size. A small number of subjects could be responsible for the reduction of the statistical power of the results we discussed. For future studies, it would also be relevant to add a control active condition engaging the episodic memory, such as a free recall of a list of words. This could help verifying the specific activation patterns of each type of long-term memory task. Moreover, a resting state condition to compare with the story recall may highlight further findings in terms of brain activation properly linked to the task.

However, the promising outcomes of this pilot study are the first step to collect fMRI data from a larger sample and from participants of different age-ranges, with a main focus on the elderly population.

Lastly, future research might use non-invasive brain stimulation techniques delivered on specific parts of the AG, such as the transcranial magnetic stimulation, in order to detect possible effects of the cerebral perturbation and connecting them with the retrieval performance.

## 5. Conclusions

The outcomes extend previous findings in the field of episodic memory investigation using an innovative free recall task during an fMRI scanning session: RSNs/IPL patterns of activation are connected to the free recall task, with the left AG being highly recruited during the effort to retrieve verbal information. Moreover, higher memory performance is connected to a balanced coactivation of the Language, Default Mode, Left executive Control and Precuneus networks and a strong involvement of the anterior part of the left AG. The data show the possibility to use a free recall paradigm during an fMRI scanning session, which could be called “imaging-supported neuropsychological examination” (I.S.N.E.). The experimental design might be used in future investigations in order to compare the observed RSNs activation of FR with brain network recruitment for others long-term memory tasks and in specific clinical populations.

## Figures and Tables

**Figure 1 brainsci-11-00911-f001:**
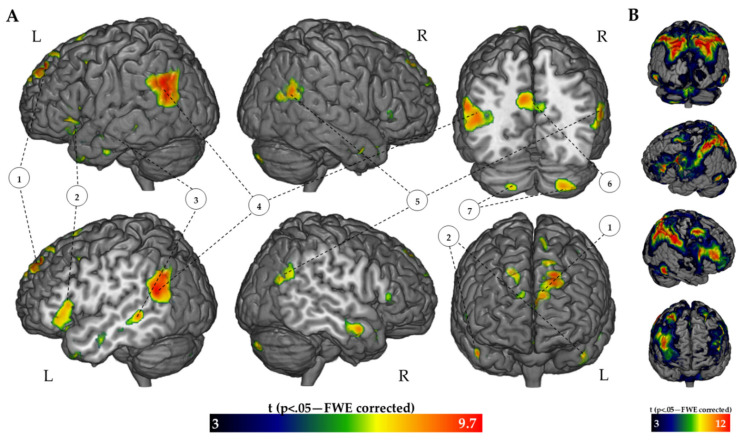
Threshold activation maps of the FR > BC contrast (**A**) and of the BC > FR contrast (**B**). The regions are displayed on the 3D rendered MNI reference brain (statistical threshold of *p* < 0.05 FWE-corrected; cluster size: >50). 1: Bilateral Superior/Medial Frontal Gyrus; 2: Left Inferior Frontal Gyrus; 3: Left Middle Temporal Gyrus 4: Left IPL; 5: Right IPL; 6: PCC and Precuneus; 7: Bilateral Cerebellum.

**Figure 2 brainsci-11-00911-f002:**
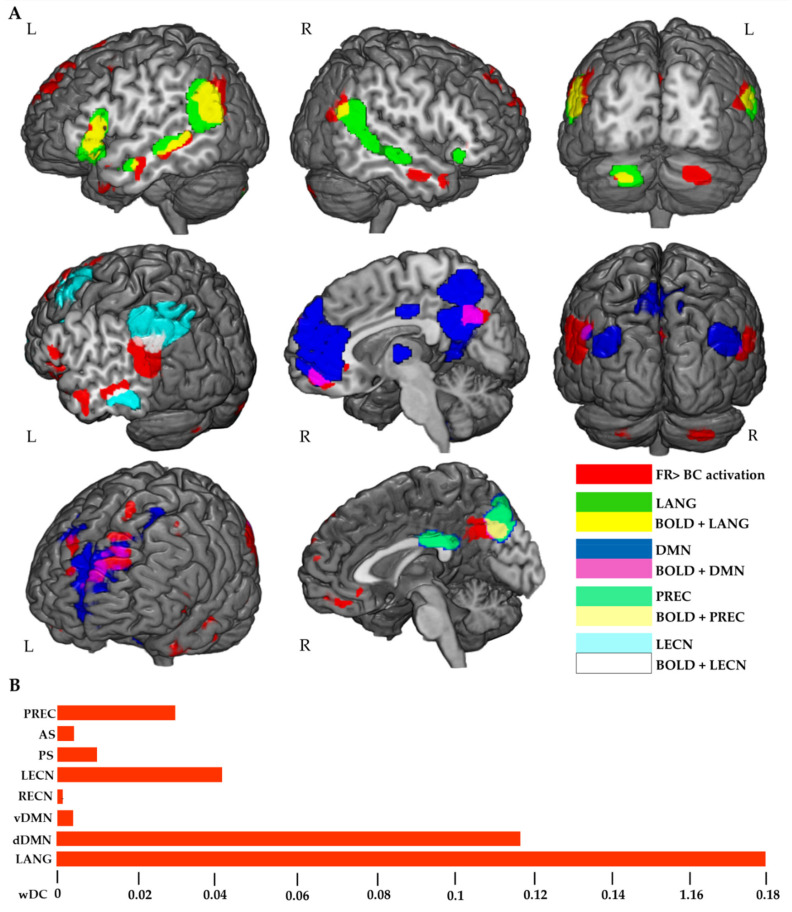
Overlap maps between FR > BC activation (RED) and RSNs. The image shows the greater similarity between activation and DMN/LANG RSNs (**A**). wDC results confirm a high similarity between FR > BC activation and dDMN, LANG, PREC and LECN networks (**B**). vDMN and dDMN: ventral and dorsal default mode network, respectively; RECN and LECN, right and left executive control networks, respectively; AS and PS, anterior and posterior salience networks, respectively; LANG, language network; PREC, precuneus network.

**Figure 3 brainsci-11-00911-f003:**
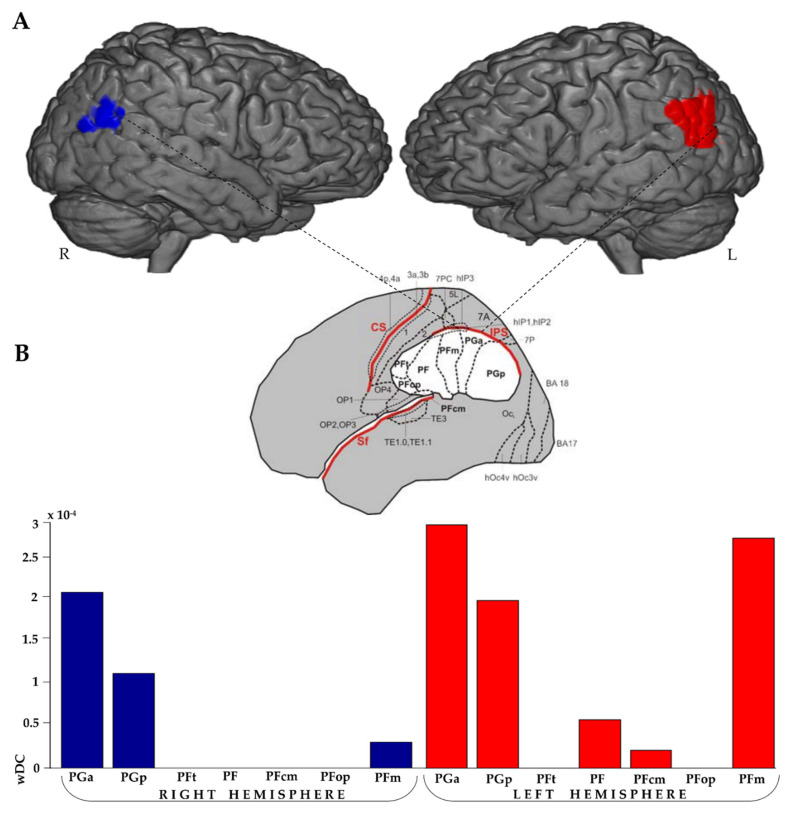
Overlap maps between FR > BC activation and IPL parcellation. Overlap pattern differences between brain hemispheres and within both left and right IPL are shown (**A**). Spatial similarity indices extracted with the wDC analysis for FR > BC contrast activation and IPL parcellation (**B**).

**Figure 4 brainsci-11-00911-f004:**
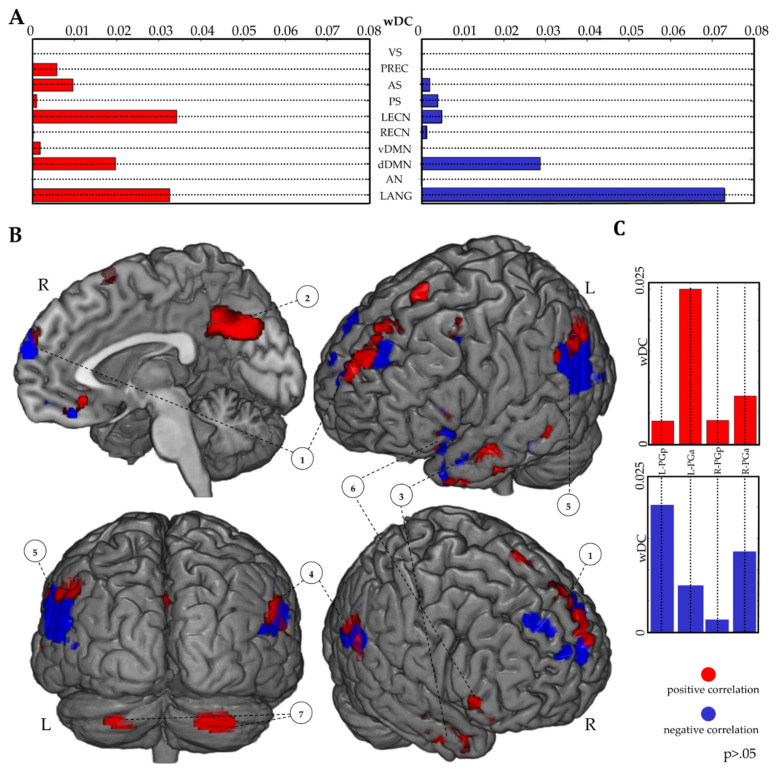
Performance-based analysis results. wDC analysis results using suprathreshold correlational maps (*p* > 0.05; FR > BC contrast; IR as a covariate of interest) and functional RSNs (**A**). Suprathreshold correlational maps (*p* > 0.05; FR > BC contrast; IR as a covariate of interest) are reported (**B**). wDC analysis results between suprathreshold correlational maps and AG of the IPL parcellation atlas (**C**). A positive correlation is shown in red; a negative correlation is shown in blue. 1: Bilateral Superior/Medial Frontal Gyrus; 2: Precuneus and PPC; 3: Left Middle Temporal Gyrus; 4: Left IPL; 5: Right IPL; 6: Left Inferior Frontal Gyrus; 7: Bilateral Cerebellum.

**Table 1 brainsci-11-00911-t001:** MNI coordinates for each region, showing an increased BOLD signal in the FR > BC contrast.

Activation Loci (FR > BC Contrast)	Cluester Size (Voxels)	t	Coordinates (MNI)		
			X	Y	Z
*Left Parietal-Occipital cortex*	1386				
Angular Gyrus		9.70	−52	−58	22
Lateral Occipital Cortex-Sup. Division		9.19	−46	−62	26
*Left Parietal-Cingulate Cortex*	957				
Precuneus		8.41	−2	−68	34
Posterior Cingulate Cortex		8.35	−10	−50	38
*Left Frontal Cortex*	518				
Frontal Pole		9.71	−18	50	44
*Left Temporal Cortex*	492				
Planum Temporale		8.09	−50	−42	0
Middle Temporal Gyrus-Posterior		7.38	−62	−40	−2
*Right Parietal Cortex*	477				
Angular Gyrus		9.37	62	−58	24
*Left Frontal Cortex*	473				
Frontorbital Cortex		8.20	−46	30	−8
Inferior Frontal Gyrus-Pars Triangularis		7.61	−52	20	8
*Left & Right Frontal Cortex*	305				
Frontal Pole		7.03	10	44	−22
Frontal Medial Cortex		6.91	8	54	−12
*Right Cerebellum*	278				
Crus II		8.04	24	−82	−38
*Right Temporal Cortex*	210				
Middle Temporal Gyrus-Ant. Division		9.07	54	4	−26
Middle Temporal Gyrus-Post. Division		5.04	58	−8	−18
*Left Temporal Cortex*	202				
Superior Temporal Gyrus-Posterior		6.94	−60	−2	−22
Medial Temporal Gyrus-Ant.		6.93	−54	−8	−22
*Right Frontal Cortex*	139				
Frontal Pole		8.55	18	48	48
*Left Frontal Cortex*	139				
Superior Frontal Gyrus		7.26	−6	16	64
*Left Temporal Cortex*	127				
Temporal Pole		7.42	−44	18	−34
*Left Frontal Cortex*	109				
Frontal Pole		6.87	−12	44	52
*Left Frontal Cortex*	107				
Middle Frontal Gyrus		6.67	−38	10	46
*Left Cerebellum*	80				
Crus II		7.16	−24	−82	−38
*Right Frontal Cortex*	72				
Frontal Pole		6.14	50	36	−4
Inferior Frontal Gyrus-Pars Triangularis		5.78	60	30	6
*Right Temporal Cortex*	71				
Temporal Pole		8.61	48	18	−32
*Right Frontal Cortex*	63				
Frontal Pole		6.02	12	62	26

## Data Availability

The datasets generated and analysed during the current study are available from corresponding author on reasonable request.

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
