# Peer review of "Brain Functional Correlates of Episodic Memory Using an Ecological Free Recall Task"

_brainsci, 2021, doi:10.3390/brainsci11070911_

Round 1

Reviewer 1 Report

Review for the article: Brain functional correlates of episodic memory using an ecological free-recall task, Authors: Francesco Neri, Stefano F. Cappa, Lucia Mencarelli, Davide Momi, Emiliano Santarnecchi, Simone Rossi.

Undoubtedly episodic memory allows us to revive a past event through mental time-travel. Authors using an ecological fMRI-free recall task, aimed to investigate the relevant recruitment of the brain networks associated with the story recollection process and its performance. Fourteen healthy participants listened to a brief story and were tested for Immediate-Recall. 

My comments to the article are as follows:

- I propose to supplement the keywords with: Episodic Memory.

- I suggest you combine the individual sections within Abstract into one large whole. Such a clear division of the Abstract into sections is not used.

- I propose to provide a detailed argumentation on what basis was this and not another research group selected?

- Have you planned to carry out data analysis with a method other than Matlab? Why did you decide to choose this particular product? How did this affect the analysis.

- Conclusions should contain plans for the future regarding the conducted research. Please transfer the description from point 4 in this respect to point 5.

Author Response

- I propose to supplement the keywords with: Episodic Memory.

RESPONSE: thanks to reviewer for the suggestion. We added the keyword “episodic memory” to the list

- I suggest you combine the individual sections within Abstract into one large whole. Such a clear division of the Abstract into sections is not used.

RESPONSE: thanks to reviewer for noticing this. We removed the headers from abstract.

- I propose to provide a detailed argumentation on what basis was this and not another research group selected?

RESPONSE: thanks to reviewer for allowing us to explain this point. Given the novelty of the paradigm we wanted to be sure of its feasibility and we selected a group of healthy young participants, before proposing it to a specific population (healthy elderly, patients with MCI or AD). We are planning to use the experimental design in other groups in order to observe BOLD signal differences or RSNs specific activations. Thanks to this and your last comment, we have now remarked this point in the conclusion section, stressing the need to employ this paradigm in different groups.

- Have you planned to carry out data analysis with a method other than Matlab? Why did you decide to choose this particular product? How did this affect the analysis.

RESPONSE: thanks to reviewer for let us explain this point. We have chosen to use the SPM package, because it is a reliable (free) open-source Matlab-based cross-platform software useful to perform standardized computation and analyzes of BOLD signal. The tool has been used and cited in thousands of studies and we believe that its use in our work increase the validity of the outcomes.

- Conclusions should contain plans for the future regarding the conducted research. Please transfer the description from point 4 in this respect to point 5.

RESPONSE: thanks to reviewer for this comment. We added a sentence in the conclusions section, where we remark the importance to conduct further investigation to explore RSNs peculiarities in specific populations: “…The experimental design might be used in future investigations in order to compare the observed RSNs activation of FR with brain networks recruitment during others long-term memory tasks execution and in specific clinical populations.…”

Reviewer 2 Report

Review – Brain Sciences: Brain functional correlates of episodic memory using an ecological free-recall task

In this study, to investigate the neural correlates of episodic memory, the authors used, for the first time, an ecological free recall task, in which participants were asked to recall a story during fMRI acquisition (vs. to perform a backward counting task). Behavioral performances were assessed right after the story was presented (Immediate recall), and after the MRI acquisition (Delayed recall). The results showed a strong correlation between the two recall phases, significant activations in the FR > BC contrast in the PPC (stronger on the left hemisphere) that overlapped strongly with the language network, the dorsal DMN, the precuneus and the LECN; and better recall associated with stronger activation in the left anterior AG compared its posterior part.  I think this paper is really well written, and constitutes an important piece of work that would advance knowledge in the field. I only have minor comments that are listed below.

First, I think the manuscript would benefit from having a more detailed figure including the different subsections of the PPC, AG, etc.  that would help the reader understand the organization of these structures (I am not really familiar with the parietal cortex and it would have helped me to have this type of figure. I realized that Figure 3 almost contains all the information but it is really small and hard to read).

In the methods section, it is mentioned that there are 30 “concepts” in each passage can the authors better explain what those concepts are, maybe just by providing an example of one of these passages?

Regarding the number of volumes acquired in the FR vs. BC. Tasks, maybe I misunderstood but the methods states that the 2 tasks alternate every 20 fMRI volumes. However, it also said that a total of 90 volumes were acquired for FR, how is this 90 and not 80 (4 repetitions) or 100 (5 repetitions), can the authors clarify?

In the results section, it is said that participants retrieved an average of 16 concepts, that is just slightly above 50% of the initially presented concepts, is this level of performance usual?

On Figure 1, why is the color bar starting at t = 0, this is not significant p-values?

More generally, I was wondering whether the results would be different if the subjects were asked to visually mentally recall these concepts – instead of subvocalizing, I guess it could be another control condition to dissociate the activation due to language vs. actual episodic memory. Finally, I found very interesting the correlation between recall quality and brain activation, I think it offers potential new targets for brain stimulation studies that could be developed in the treatment of Alzheimer disease for example.

Author Response

- First, I think the manuscript would benefit from having a more detailed figure including the different subsections of the PPC, AG, etc.  that would help the reader understand the organization of these structures (I am not really familiar with the parietal cortex and it would have helped me to have this type of figure. I realized that Figure 3 almost contains all the information but it is really small and hard to read).

RESPONSE: thanks to reviewer for this helpful comment. We changed the figure 3, scaling down the bar chart and enlarging the part representing the divisions of the parietal lobe by Caspers and colleagues. We hope that the reader can now quickly view the subdivisions of the angular / supramarginal gyrus that we report in the paper.

- In the methods section, it is mentioned that there are 30 “concepts” in each passage can the authors better explain what those concepts are, maybe just by providing an example of one of these passages?

RESPONSE: we thank the reviewer for giving us the opportunity to clarify the point. We have expanded the supplementary material and created a specific methods section in order to report one of the passages used in the study both in Italian and English languages.

- Regarding the number of volumes acquired in the FR vs. BC. Tasks, maybe I misunderstood but the methods states that the 2 tasks alternate every 20 fMRI volumes. However, it also said that a total of 90 volumes were acquired for FR, how is this 90 and not 80 (4 repetitions) or 100 (5 repetitions), can the authors clarify?

RESPONSE: we really thank the reviewer for noticing this, allowing us to clarify this point. The scan consists of 178 volumes. 3 volumes are removed for the steady-state magnetization process, as described in the manuscript. The 175 remaining volumes were assigned to the two tasks as follows:

1-20: FR task;

21-40: BC task;

41-60: FR task;

61-80: BC task;

81-100: FR task;

101-120: BC task;

121-140: FR task;

141-160: BC task;

161-175: FR task.

Total volumes for FR: 95

Total volumes for BC: 80

The typo has been corrected in the manuscript.

- In the results section, it is said that participants retrieved an average of 16 concepts, that is just slightly above 50% of the initially presented concepts, is this level of performance usual?

RESPONSE: thanks to reviewer for let us clarifying this point. In order to correlate the scores with the neuroimaging data we wanted to be sure that the participants would reach a performance quite far from 100%. In literature, we found that the mean performance of the free recall in healthy young people varies accordingly to the length of the passage. In a study by Carlesimo and colleagues (Carlesimo et al., 2002), a very-short story was standardized on a sample of healthy young subjects. It was found out that the subjects were able to recall an average of 5.9 elements out of 8 (73%). In 1987 Barigazzi and colleagues used a longer prose passage (174 words / 51 concepts to be recalled). In this case, a subsample of 10 healthy young subjects were able to recall an average 51% of concepts (Barigazzi et al., 1987), performance that is comparable to our study results. We now report this information in the discussion section:”...The IR/DR performance (~56% of correct concepts recalled) is comparable to previous findings: in 1987 Barigazzi and colleagues collected normative data using a prose passage (174 words / 51 concepts). In this case, a subsample of 10 healthy young subjects recalled on average the 51% of total concepts [53]…”

- On Figure 1, why is the color bar starting at t = 0, this is not significant p-values?

RESPONSE: we thank the reviewer for noting this. The value reported in the colorbar was a typo. This has been changed with the real initial value: 3. The figure has been inserted again with the correct value.

- More generally, I was wondering whether the results would be different if the subjects were asked to visually mentally recall these concepts – instead of subvocalizing, I guess it could be another control condition to dissociate the activation due to language vs. actual episodic memory. Finally, I found very interesting the correlation between recall quality and brain activation, I think it offers potential new targets for brain stimulation studies that could be developed in the treatment of Alzheimer disease for example.

RESPONSE: dear reviewer, we also tried to correlate for other questionnaire scores, such as that of the visual imagination result. Unfortunately, we did not find any other differences to report in the manuscript. We speculate that with a larger sample size and specific clinic populations is possible to obtain other interesting correlations.

Reviewer 3 Report

The study assessed hemodynamic activity during retrieval of verbal episodic memories formed immediately before the fMRI scan (short passage). In a block-design framework they were asked to silently retrieve the studied passage during 4 (?) 50 sec blocks alternating with a subtraction task. 

Whole-brain GLM was used to determine regions that show relatively increased BOLD during the retrieval blocks and vice versa. ROI analyses were also conducted to identify parietal sites predominantly activated during the retrieval blocks.

Whereas the logic of the analyses and the research hypotheses are sound and interesting, the design of the fMRI task is problematic. 

Specifically, I don't see how the whole-brain activity maps revealed in the memory>subtraction comparison can be attributed to episodic retrieval, given that the comparison condition is so vastly different from the task the subjects were asked to engage in during the "memory" blocks. 

Secondly, the second research question would best be served by MVPA approaches which are data-driven and sensitive to subtle individual differences in the spatial layout of activation profiles.

Author Response

RESPONSE: dear reviewer, thank you for your constructive comment that has given us the possibility to critically reflect on our results. 

Regarding the first question, we have initially planned to use a resting state condition to compare with FR task, but we did not want to let the participants to engage the memory task involuntary. Moreover, we planned to compare the free recall with others long-term memory task, but we were concerned that this might create an interfering effect between verbal information to be recalled.

The BC task was chosen in order to constantly engage the subjects’ attentive resources in an active and quite-challenging task and to decrease the risk that participants would think to the prose passage during the control task.

The activation that we found in the BC>FR comparison is comparable with previous findings where the task has been contrasted with a resting-state condition (Yarets et al., 2019) and the RSNs recruitment of FR>BC is similar to that of previous findings, where verbal long-term memory was investigated, but using different memory tasks (Cabeza et al., 2008; Bonnici et al., 2016; Chen H-Y et al., 2017; Chen J. et al., 2017).

The BC task has some characteristics in common with FR task: evocation of verbal information in a silent way and other far from it, as demonstrate by the DAN recruitment during the BC only. In fact, the BC task requires retrieving the subtraction rules from the long-term memory and applying them to perform the arithmetic operation, therefore drawing upon the central executive. Moreover, during mental arithmetic tasks people need to hold the intermediate products which tend to use the phonological loop (Seitz & Schumann Hengsteler, 2002; Yang, 2011).

However, in order to explore if specific brain patterns depend on the material to retrieve, we are planning to compare the FR task activation with other types of ecological long-term memory tasks, such as a visuospatial memory task or a face recognition task and we have remarked this point in the conclusions section.

References:

Bonnici HM, Richter FR, Yazar Y, Simons JS. Multimodal Feature Integration in the Angular Gyrus during Episodic and Semantic Retrieval. J Neurosci 2016;36:5462–71. https://doi.org/10.1523/JNEUROSCI.4310-15.2016.

Cabeza R, Ciaramelli E, Olson IR, Moscovitch M. The parietal cortex and episodic memory: an attentional account. Nat Rev Neurosci 2008;9:613–25. https://doi.org/10.1038/nrn2459.

Chen H-Y, Gilmore AW, Nelson SM, McDermott KB. Are There Multiple Kinds of Episodic Memory? An fMRI Investigation Comparing Autobiographical and Recognition Memory Tasks. J Neurosci 2017;37:2764–75. https://doi.org/10.1523/JNEUROSCI.1534-16.2017.

Chen, J., Leong, Y., Honey, C. et al. Shared memories reveal shared structure in neural activity across individuals. Nat Neurosci 20, 115–125 (2017).

Seitz, K., & Schumann-Hengsteler, R. (2002). Phonological loop and central executive processes in mental addition and multiplication. Psychologische Beitrage, 44(2), 275–302.

Yarets MY, Sharova EV, Smirnov AS, Pogozbekyan AL, Boldyreva GN, Zaytsev OS, et al. Analysis of the Structural-Functional Organization of a Counting Task in the Context of a Study of Executive Functions. Neurosci Behav Physiol 2019;49:694–703. https://doi.org/10.1007/s11055-019-00789-x.

RESPONSE: Regarding the second question, we really thank you to give us the opportunity to explore our data in other directions. For this reason, we added an additional analysis based on MVPA. Here, we report what we wrote in the main manuscript as follows: “…MVPA analysis. In order to unveil for subtle patterns of activation, we carried out a multivariate pattern analysis (MVPA) using CONN toolbox. We compared FR and BC tasks and performed the analysis by retaining 2 components and. This choice was motivated by the assumption that the number of components should be equal to 10–20% of the sample size (here between 1.4 and 2.80). We show the MVPA results in the specific section supplementary material (see Figure S2)…”

The analysis that we have carried out confirmed our GLM outcome, as we find a pattern of activation similar to that we found with the previous analysis.

Thus, we decided to put the resulting figure of the MVPA analysis and the table of activation loci in the supplementary materials, citing it in the main manuscript, but still leaving the structure of the manuscript with our GLM analysis.

Reviewer 4 Report

The manuscript describes a study analyzing cerebral activations during covert recall of a story contrasted to backward counting in a small sample of healthy participants. Story recall elicited activation in several regions from the language, default mode and left executive networks.

The manuscript is well-written and clear. The topic is of interest, but the method lacks of control, inherently because of the covert recall protocol. I detail below my main concerns relative to the study.

  1. In the abstract, the authors claim that no neuroimaging study has used an ecological task such as a recall of a structured story. However, there have been several publications looking at the neural bases of recall of naturalistic events, such as recalling the story from a movie. In the introduction, the authors actually refer to Chen et al. (2017) who studied neural correlates of movie recall. This type of work, as well as the work by Uri Hasson and his colleagues more generally, could further be used to formulate a priori hypotheses for the current study, as their approach has similarities with story recall.

  1. Lines 109-111: The goal of the study is “to detect brain activation during verbal recollection, identify which subregions of the ventral PPC or IPL are functionally recruited and verify if specific IPL areas or brain networks might particularly contribute to a better EM performance”. It is not clear why the focus is on parietal areas, as other areas have also been put forward as critical for episodic memory in naturalistic contexts (e.g., hippocampus, posterior cingulate/retrosplenial cortex). Some argumentation for the focus on parietal areas is needed. Also what hypothesis about the role of the parietal cortex is tested in this study?

  1. Lines 140-146: If I understood well the methods, each participant only studied and recall 1 story. In alternated blocks, the participant had to recall repeatedly the same story. The number of blocks is not specified, but I guess it is around 8 blocks in total based on the number of volumes. If a participant had to recall the same story 4 times, it is possible that, with repetition, what is recalled is more and more schematized. Indeed, repetition tends to promote semantisation and loss of specific details (e.g., Reagh & Yassa, 2014, Learning and Memory). So, how can we be sure that the fMRI sessions measured episodic memory and not semantic memory? The high subjective rating of similarity of free recall with immediate and delayed recall does not guarantee that the information that participants rated is episodic in nature. Indeed, participants could say that their recall is very similar because they had a consistent memory for the themes and concepts from the story.

  1. Another potential difficulty with the lack of control over what is actually recalled in the scanner is the notion of temporal compression. When we mentally relive a past event, we tend to relive it much more quickly than it lasted in reality. So it is likely that the covert recall of the story did not occupy the 50-second period, leaving room for mind-wandering (which is usually associated with the same brain areas as episodic memory).

  1. Lines 163: What was the field strength of the scanner (3T?)?

  1. Lines 305-306: The authors claim that their task was “simulating an ecological communicative situation, where a person has to recall a story in order to share it with someone else” However, there is no evidence in the methods that participants were trying to communicate the story to another person, nor even simulating this transmission to another person. Instead, the task involved covert private recall.

Author Response

1. In the abstract, the authors claim that no neuroimaging study has used an ecological task such as a recall of a structured story. However, there have been several publications looking at the neural bases of recall of naturalistic events, such as recalling the story from a movie. In the introduction, the authors actually refer to Chen et al. (2017) who studied neural correlates of movie recall. This type of work, as well as the work by Uri Hasson and his colleagues more generally, could further be used to formulate a priori hypotheses for the current study, as their approach has similarities with story recall.

RESPONSE: we thank the reviewer for the suggestion. We have now inserted the citation of Uri Asson and colleagues in the introduction section, that is a valid support to our paradigm: “...In neuroimaging studies, a free recall of a structured event has been rarely adopted as a protocol to investigate the neural basis of EM retrieval process. fMRI activity has been measured while participants encode or retrieve a TV show or a movie[17,18]. More, other authors have preferred to use an autobiographical memory task…”

2. Lines 109-111: The goal of the study is “to detect brain activation during verbal recollection, identify which subregions of the ventral PPC or IPL are functionally recruited and verify if specific IPL areas or brain networks might particularly contribute to a better EM performance”. It is not clear why the focus is on parietal areas, as other areas have also been put forward as critical for episodic memory in naturalistic contexts (e.g., hippocampus, posterior cingulate/retrosplenial cortex). Some argumentation for the focus on parietal areas is needed. Also what hypothesis about the role of the parietal cortex is tested in this study?

RESPONSE: thanks to reviewer for asking this. The role of the retrosplenial cortex, posterior cingulate cortex and hippocampus in memory functions is well established as well as the pivotal role of parietal cortex. Our long-term aim is to modulate memory function using non-invasive brain stimulation (NIBS) on a fundamental brain node of memory function. We have chosen to focus on the parietal cortex because it is easily accessible through NIBS techniques and we are planning to stimulate different parts of the angular gyrus in order to verify the effects on the story performance. In fact, the retrosplenial cortex and cingulate cortex appear to be more difficult to stimulate than the parietal areas. Wang's study (Wang et al., 2014) confirms that the parietal is one of the fundamental nodes (together with retrosplenial and PPC) of the memory system and that through its stimulation the hippocampus can also be indirectly reached.  We added a paragraph in the “Limitations and future directions” section, accordingly: “…Lastly, future researches might use non-invasive brain stimulation techniques delivered on specific parts of the AG, such as the transcranial magnetic stimulation, in order to detect possible effects of the cerebral perturbation and connecting them with the retrieval performance…”

References:

Wang, J. X., Rogers, L. M., Gross, E. Z., Ryals, A. J., Dokucu, M. E., Brandstatt, K. L., Hermiller, M. S., & Voss, J. L. (2014). Targeted enhancement of cortical-hippocampal brain networks and associative memory. Science (New York, N.Y.)345(6200), 1054–1057. https://doi.org/10.1126/science.1252900.

3. Lines 140-146: If I understood well the methods, each participant only studied and recall 1 story. In alternated blocks, the participant had to recall repeatedly the same story. The number of blocks is not specified, but I guess it is around 8 blocks in total based on the number of volumes. If a participant had to recall the same story 4 times, it is possible that, with repetition, what is recalled is more and more schematized. Indeed, repetition tends to promote semantisation and loss of specific details (e.g., Reagh & Yassa, 2014, Learning and Memory). So, how can we be sure that the fMRI sessions measured episodic memory and not semantic memory? The high subjective rating of similarity of free recall with immediate and delayed recall does not guarantee that the information that participants rated is episodic in nature. Indeed, participants could say that their recall is very similar because they had a consistent memory for the themes and concepts from the story.

RESPONSE: we really thank the reviewer for letting us to clarify and critically reflect on this point.

As you rightly say, we can’t guarantee that the participants’ nature of recall during FR might be equal to that they had during IR or DR, for the absence of a performance during the scanner session. We were only able to verify the similarity recall with a questionnaire only.

Regarding the semanticization caused by repetition, we now report in the method section a detailed description of the instructions that were given to participants, trying to take advantage of your valuable comment: “…To avoid a schematization and a semanticization of the information caused by the multiple repetitions inside and outside the scanner, we instructed the participants to keep the recall as more similar as possible…”

The quantity of information between the two free recalls remained unchanged, including details reported and we did not observe a schematization of the recall in delayed recall. Moreover, an high correlation between immediate and delayed recall is not novel, even in absence of intermediate repetitions between immediate and delayed recall. In the manuscript two example are already reported, but now we provide two more (studies by Carlesimo et al., 2002 and by Barigazzi et al., 1987), that found an high correlation between recalls.

4. Another potential difficulty with the lack of control over what is actually recalled in the scanner is the notion of temporal compression. When we mentally relive a past event, we tend to relive it much more quickly than it lasted in reality. So it is likely that the covert recall of the story did not occupy the 50-second period, leaving room for mind-wandering (which is usually associated with the same brain areas as episodic memory).

RESPONSE: we thank the reviewer for letting us to describe our paradigm in detail. We pointed out more in the methods section: “…The mind-wondering was not allowed and participants had to perform the same task for the entire block. If they finished to repeat the passage during FR or they reached 0 during the BC, they had to start again the task until the experimenter’s signal…”

5. Lines 163: What was the field strength of the scanner (3T?)?

RESPONSE: the scanner was a 1.5 T. We specified this in methods section. Many thanks to let us to precise this.

6. Lines 305-306: The authors claim that their task was “simulating an ecological communicative situation, where a person has to recall a story in order to share it with someone else” However, there is no evidence in the methods that participants were trying to communicate the story to another person, nor even simulating this transmission to another person. Instead, the task involved covert private recall.

RESPONSE: thanks to reviewer for let us think about this point.  Indeed, it may be too speculative to state this and we decided to change the sentence accordingly to your comment: “…Our aim was to load continuously the “recall cognitive function” during FR fMRI blocks acquisition using non-autobiographical memories and simulating an ecological situation, where a person has to recall a story in order remember it as well as possible…”

Round 2

Reviewer 3 Report

The authors responded adequately to the comments on the earlier version of the MS

Reviewer 4 Report

The revision has clarified most of my concerns.